# Tracking zoonotic pathogens using blood-sucking flies as 'flying syringes'

Paul-Yannick Bitome-Essono[1,2,3]*, Benjamin Ollomo[2†], Céline Arnathau[4], Patrick Durand[4], Nancy Diamella Mokoudoum[2], Lauriane Yacka-Mouele[2], Alain-Prince Okouga[2], Larson Boundenga[2], Bertrand Mve-Ondo[2], Judicaël Obame-Nkoghe[2], Philippe Mbehang-Nguema[2,3], Flobert Njiokou[5], Boris Makanga[2,3], Rémi Wattier[1], Diego Ayala[2,4], Francisco J Ayala[6], Francois Renaud[4], Virginie Rougeron[2,4], Francois Bretagnolle[1†], Franck Prugnolle[2,4*†], Christophe Paupy[2,4*†]

[1]Biogéosciences Unit, Équipe Écologie-Évolutive, UMR 6282 CNRS-université de Bourgogne-Franche Comté-EPHE-AgroSup, Dijon, France; [2]Équipes UBEEP-ESV, Centre International de Recherches Médicales de Franceville, Franceville, Gabon; [3]Département de Biologie et Écologie Animale, Institut de Recherche en Écologie Tropicale, Libreville, Gabon; [4]MIVEGEC Unit, UMR 224-5290 IRD-CNRS-UM, Centre IRD de Montpellier, Montpellier, France; [5]Département de Biologie Animale et Physiologie, Laboratoire de Parasitologie et Écologie, Faculté des Sciences de l'Université de Yaoundé 1, Yaoundé, Cameroun; [6]Department of Ecology and Evolutionary Biology, University of California, Irvine, United States

*For correspondence:
bitomessono@yahoo.fr (P-YB-E);
franck.prugnolle@ird.fr (FP);
christophe.paupy@ird.fr (CP)

†These authors contributed equally to this work

Competing interests: The authors declare that no competing interests exist.

**Abstract** About 60% of emerging infectious diseases in humans are of zoonotic origin. Their increasing number requires the development of new methods for early detection and monitoring of infectious agents in wildlife. Here, we investigated whether blood meals from hematophagous flies could be used to identify the infectious agents circulating in wild vertebrates. To this aim, 1230 blood-engorged flies were caught in the forests of Gabon. Identified blood meals (30%) were from 20 vertebrate species including mammals, birds and reptiles. Among them, 9% were infected by different extant malaria parasites among which some belonged to known parasite species, others to new parasite species or to parasite lineages for which only the vector was known. This study demonstrates that using hematophagous flies as 'flying syringes' constitutes an interesting approach to investigate blood-borne pathogen diversity in wild vertebrates and could be used as an early detection tool of zoonotic pathogens.

## Introduction

Emerging and re-emerging human infectious diseases have increased in recent years. Around one-fourth of the 1415 pathogens known to infect humans appeared between 1940 and 2004 and their appearance has gradually increased since 1980 (*Taylor et al., 2001*; *Woolhouse and Gaunt, 2007*; *Jones et al., 2008*; *Daszak et al., 2004*). Today, seven new pathogens appear every year and this number should reach 15–20 by 2020 (*Woolhouse et al., 2008*), mostly due to the growth of human activities that increase contact with novel sources of pathogens and favor their spread worldwide (*Murray et al., 2015*). Emerging threats mainly concern viruses, such as HIV (*Sharp and Hahn, 2011*), SARS-CoV and MERS-CoV (*de Wit et al., 2016*), avian flu (*Alexander, 2007*) and more recently Ebola (*Baize et al., 2014*), chikungunya (*Burt et al., 2012*) and Zika (*Wikan and Smith, 2016*). However, disease emergence and re-emergence also concern bacteria (e.g. *Helicobacter*

**eLife digest** About 60% of new infectious diseases in humans come from animals. Their increasing number and rapid spread are linked to increasing levels of contact between humans and wildlife, as recently highlighted by the epidemics of Zika in Brazil or Ebola in West Africa. To anticipate and prevent similar outbreaks in the future, it would be ideal to develop new methods for the early detection and monitoring of infectious diseases in wild animals.

Currently, three methods are mainly used to screen wild animals for infectious disease, but these all have limitations. Analyses of bushmeat and game meat only investigate those animals that are eaten by humans. Testing the organs and tissues of trapped animals can be difficult and harmful for both the humans and animals involved. Collecting and examining samples of feces, urine or saliva cannot detect all diseases and can be difficult to do for some species.

Bitome-Essono et al. now demonstrate a new method for assessing the diseases carried by wild animals: using blood-sucking flies as 'flying syringes' to collect their blood. During several weeks of sampling in Gabon, Central Africa, Bitome-Essono et al. trapped thousands of these flies, about a third of which were engorged with blood. Analyses of these blood samples revealed that they had come from 20 different species, including birds, mammals and reptiles. Different malaria parasites could also be detected in the blood.

Although the study performed by Bitome-Essono et al. only focused on malaria parasites, in the future the technique could be extended to analyze a number of disease-causing microbes – including viruses, bacteria, protozoa and macroparasites – that are found in the blood of wild animals.

*pylori, Salmonella* sp., etc.) and parasites (e.g. *Plasmodium knowlesi* in South-East Asia). Sixty per cent of diseases emerging in humans are zoonoses and wildlife plays a key role by providing a zoonotic pool from which previously unknown pathogens may emerge (*Taylor et al., 2001*; *Woolhouse and Gaunt, 2007*; *Jones et al., 2008*; *Daszak et al., 2004*). The case of *P. knowlesi* in South-East Asia is a good example. This parasite emerged in the human population after a transfer from Asian macaques. It is now considered as the fifth human malaria agent after *Plasmodium falciparum*, *Plasmodium vivax*, *Plasmodium malariae* and *Plasmodium ovale* (*Singh and Daneshvar, 2013*). Such emerging diseases constitute a massive public health issue that requires active monitoring for signs of outbreaks and rapid diagnosis of the involved pathogen. Therefore, it is crucial to anticipate and prevent potential epidemic and pandemic outbreaks by developing new methods for the early detection and monitoring of infectious agents in wild animal sources (*Kuiken et al., 2005*; *Wolfe et al., 2005*). However, in many cases, monitoring is limited or impossible due to our poor knowledge about the ecology of these pathogens (i.e. where, when and how these agents circulate in the wildlife). The case of the Ebola virus is quite exemplary. Indeed, the exact nature of its reservoir(s) remains uncertain, although thousands of animals have been screened during the last 40 years (e.g. [*Marí Saéz et al., 2015*]).

Nowadays, pathogen circulation in wild animals is screened using mainly two methods: bushmeat analysis or direct trapping of animals for organ and tissue collection. These methods are pertinent in many cases, but present some weaknesses. Bushmeat represents only a fraction of the fauna (the one consumed by humans), whereas animal trapping can be difficult or dangerous. Moreover, such manipulation may be harmful for threatened and protected species. As a consequence, several methods were developed in the last years to study pathogen diversity from wild fauna without the need of direct contacts with animals, for example, by using fecal, urine or saliva samples (e.g. [*Santiago et al., 2002*; *Prugnolle et al., 2010*; *Pesapane et al., 2013*; *Taberlet et al., 2012*]). However, the value of these non-invasive methods remains limited because not all pathogens can be detected and not all reservoirs can be explored by these methods (for instance, it is difficult to collect feces or saliva of reptiles without trapping them). Therefore, new non-invasive methods are crucially needed to provide new opportunities for screening a larger range of hosts and pathogens.

The use of hematophagous flies as 'flying syringes' may constitute a new approach to track and survey blood-borne pathogens in the wild (*Calvignac-Spencer et al., 2013*). Nucleic acids (DNA or

RNA) of vertebrate hosts or of pathogens in arthropod blood meals are preserved and detectable for several days (*Calvignac-Spencer et al., 2013*; *Kent, 2009*; *Muturi et al., 2011*; *Grubaugh et al., 2015*; *Lee et al., 2015*). For example, HIV was detected 8 days and 10 to 14 days after blood ingestion by bugs and by ticks, respectively (*Webb et al., 1989*; *Humphery-Smith et al., 1993*). Recently, the H5N1 flu virus was found viable in mosquitoes (*Barbazan et al., 2008*), although its transmission by these insects is unproven (*Sawabe et al., 2006*). Grubaugh and colleagues (*Grubaugh et al., 2015*) applied such an idea (that they called 'xenosurveillance') using *Anohpeles* mosquitoes to estimate the diversity of viruses infecting human populations in remote areas. Nevertheless, blood-engorged mosquitoes are very difficult to collect in forest and often show strong host preferences (in particular for mammals). Arthropods with more generalist blood feeding patterns would be more useful to survey pathogens from a large range of vertebrates (including mammals, birds and reptiles) in these highly complex ecosystems.

Hematophagous flies (tsetse flies, stomoxids and tabanids) could be good candidates for this purpose since they are usually large Diptera (length comprised between 3 and 25 mm) and hematophagous in both sexes, with the exception of male tabanids (*Mullens, 2002*). They are easy to trap and some studies performed on tsetse flies and stomoxids showed that 20 to 40% of trapped flies are engorged with blood (*Mavoungou et al., 2008*; *Simo et al., 2012*). These flies feed on a large spectrum of vertebrate hosts, including birds, reptiles and mammals (*Muturi et al., 2011*; *Clausen et al., 1998*; *Muzari et al., 2010*). The omnipresence of hematophagous flies in certain habitats and their opportunistic blood-feeding behaviour (*Muturi et al., 2011*; *Muzari et al., 2010*; *Späth, 2000*) make of them compelling candidates to obtain blood meals from different vertebrate hosts for pathogen detection.

In the present study, we investigated the possibility of using hematophagous flies as 'flying syringes' to explore the diversity of extant malaria parasites (Haemosporida) infecting wild vertebrates living in the forests of Gabon (Central Africa).

## Results

### Host identification from blood meals

A total of 4099 hematophagous flies were caught in four national parks of Gabon during dry and rainy seasons over a cumulated sampling period of 16 weeks (*Figure 1a*). Among them, six tsetse fly species, six stomoxid species and six tabanid species were identified (*Table 1*).

Among the 4099 caught flies, 1230 (30%) were engorged with blood. These were mostly tsetse flies (n = 1218; 99%), particularly *Glossina palpalis palpalis* (n = 662; 54%) and *G. fuscipes fuscipes* (n = 214; 18%) specimens. The blood meal origin was successfully identified in 33% and 43% of these flies, respectively (*Table 1*).

Overall, the blood meal origin was successfully identified in 428 fly samples (35%) using a PCR system amplifying long fragments of *Cytb* (450 bp) or *COI* genes (330 bp or 660 bp). Specifically, blood meals were from 20 vertebrate species, including 12 families and 8 orders (*Figure 1b* and *Tables 2* and *3*).

A trial study using a PCR system amplifying a shorter fragment (150 bp of the gene 16S) to deal with potential DNA degradation in the blood meal showed a high gain of sensitivity in the determination of the origin of the blood meal. Thus, out of 89 previously unidentified blood meals, the host was identified for 76% (n = 68) of them. The list of newly identified hosts is given in *Figure 2*. This shows a high gain of sensitivity with the new PCR system.

### Pathogen identification from blood meals

Extant malaria parasites were detected in 37 (8.7%) of the 428 identified blood meals (*Figure 1c*, red isolates). Phylogenetic analyses revealed that 29.7% of these parasites belonged to *Plasmodium falciparum* (n = 11, *Figure 1c*; group 1), 8.1% to *Plasmodium adleri* (n = 3, *Figure 1c*; group 2), and 8.1% to a recently described lineage of parasites infecting wild ungulates (n = 4, *Figure 1c*; group 3) (*Boundenga et al., 2016*). For all blood meals, the identified host represented the known natural host (or one of the hosts) of such parasites. Sequences of unknown parasite lineages or of parasites for which the hosts were not known were also obtained. For instance, one sequence (*Figure 1c*; group 4) detected in a blood meal originating from an ungulate was related to parasites previously

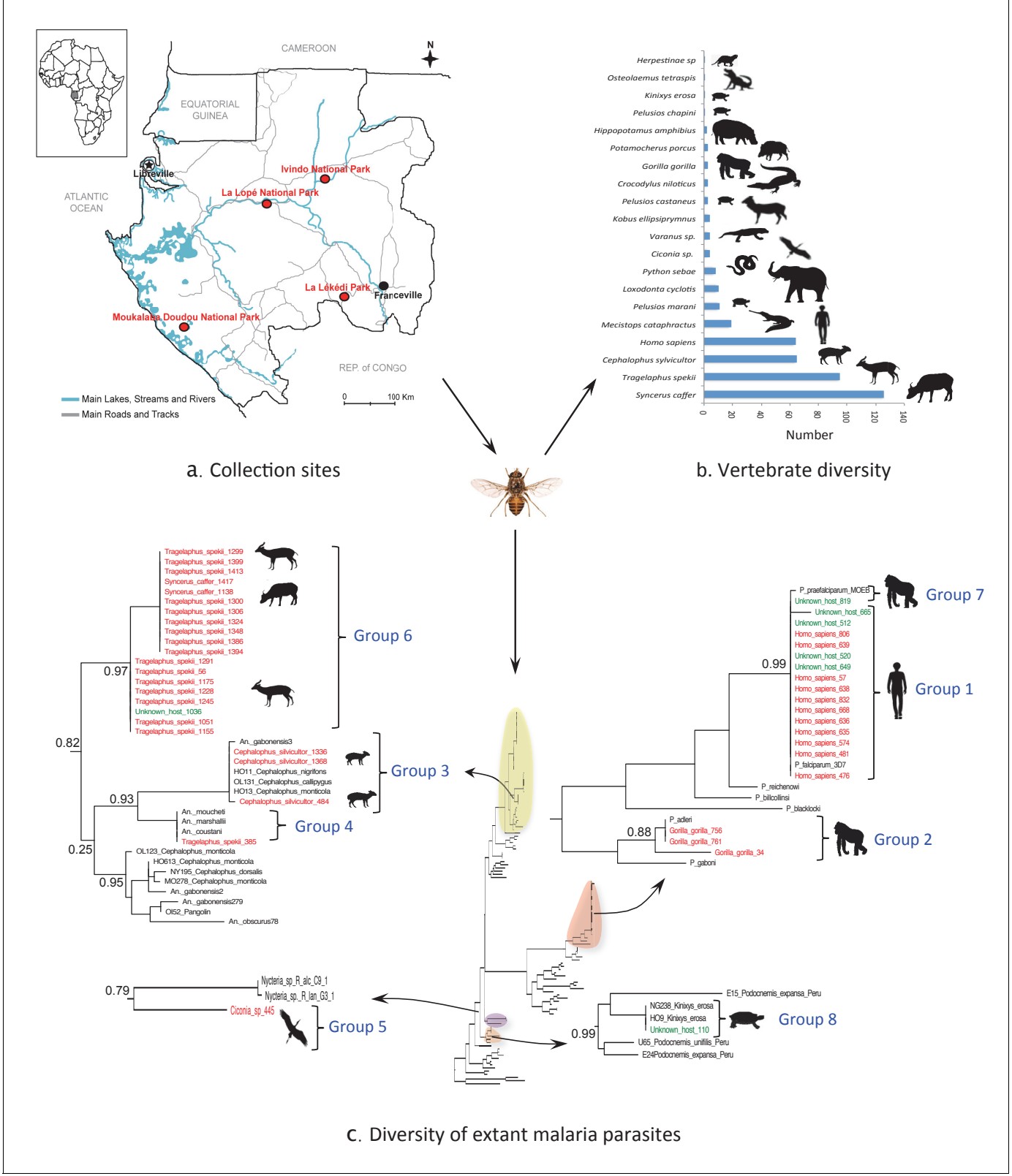

**Figure 1.** Monitoring vertebrate haemosporidian diversity using haematophagous flies. (**a**) Localization of the sampling sites (red dots) in Gabon (Central Africa). (**b**) Number of blood meals originating from the different vertebrate species. (**c**) Position within the *Cytb* phylogeny of the haemosporidian *Cytb* sequences PCR-amplified from the blood meals of engorged flies with identified hosts (red isolates) and unidentified hosts (green isolates). Black isolates: references (***Table 4***). Bootstrap values at important nodes are shown.

**Table 1.** Number and proportion of specimens captured per fly species. The number of engorged specimens and blood meals identified in each fly species are also indicated.

| Fly species | Number of collected specimens | Proportion (%) | Number of engorged specimens | Number of identified blood meals |
|---|---|---|---|---|
| Glossinidae | 2252 | 54.94 | 1218 | 423 |
| *Glossina caliginea* | 144 | 3.51 | 87 | 33 |
| *G. fusca congolensis* | 210 | 5.12 | 104 | 42 |
| *G. fuscipes fuscipes* | 290 | 7.07 | 214 | 93 |
| *G. pallicera newsteadi* | 157 | 3.83 | 97 | 37 |
| *G. palpalis palpalis* | 1372 | 33.47 | 662 | 218 |
| *G. tabaniformis* | 79 | 1.93 | 54 | 0 |
| Muscidae | 1362 | 33.23 | 9 | 4 |
| *Stomoxys calcitrans* | 245 | 5.98 | 5 | 2 |
| *S. inornatus* | 334 | 8.14 | 0 | 0 |
| *S. niger niger* | 253 | 6.17 | 4 | 2 |
| *S. niger bilineatus* | 224 | 5.46 | 0 | 0 |
| *S. omega omega* | 197 | 4.81 | 0 | 0 |
| *S. transvittatus* | 109 | 2.66 | 0 | 0 |
| Tabanidae | 485 | 11.83 | 3 | 1 |
| *Ancala sp* | 41 | 1 | 0 | 0 |
| *Atylotus sp* | 104 | 2.53 | 0 | 0 |
| *Chrysops sp* | 156 | 3.81 | 3 | 1 |
| *Haematopota sp* | 13 | 0.31 | 0 | 0 |
| *Tabanus par* | 52 | 1.27 | 0 | 0 |
| *Tabanus taeniola* | 120 | 2.93 | 0 | 0 |
| Total | 4099 | 100 | 1230 | 428 |

isolated only from *Anopheles* mosquitoes (*Boundenga et al., 2016*). One sequence detected in a blood meal originating from a bird was related to bat Haemosporida (*Nycteria*), (*Figure 1c*; group 5). Finally, 18 sequences (*Figure 1c*; group 6) that were amplified from blood meals originating from ungulates formed an independent and never described lineage related to groups 3 and 4.

In addition, 100 additional samples for which identification of the blood meal failed were randomly chosen for malarial parasite screening. This analysis showed that 7% were infected with *P. falciparum* (n = 4, group 1), *P. praefalciparum* (n = 1, group 7), malaria parasites of antelopes from group 6 (n = 1) and parasites of tortoises (group 8, n = 1) (*Figure 1c*, green isolates).

For the parasite, the use of a shorter PCR system led to less conclusive results than those obtained for the host identification. Out of the 91 blood meals that were negative to *Plasmodium* with a PCR system amplifying a long *Cytb* fragment, only one was found positive with the new system. The positive individual corresponded to a *Tragelaphus spekii* and was infected with a parasite belonging to group 3 (*Figure 1c*).

## Discussion

In this study, we tested whether hematophagous flies could be used as 'flying syringes' to identify blood-borne pathogens circulating in the wild vertebrate fauna of Gabon. Our results show that the blood meals of the captured engorged flies can be successfully used to analyze the diversity of extant malaria parasites. Despite a limited sampling effort (a total of 4 weeks of sampling for each park), we could screen the diversity of haemosporidian parasites from a large range of vertebrate hosts, including mammals, birds and reptiles. Parasites were detected in more than 8% of the analyzed samples. These malaria parasites belonged to already known, but also to never previously

**Table 2.** Number and origin of blood meals according to the fly species (Fsp), park and climatic season.

| | | | Number of identified blood meals by fly species (Fsp) | | | | | | | | | | | | | | | | | | | | | | |
|---|---|---|---|---|---|---|---|---|---|---|---|---|---|---|---|---|---|---|---|---|---|---|---|---|---|
| | | | Moukalaba-Doudou | | | | | | | | | | | Lopé | | | | | | | | | | | |
| | | | Rainy season | | | | | Dry season | | | | | | Rainy season | | | | | Dry season | | | | | | |
| Taxonomic group/ Order/Family | Host species | N° Identified | Fsp1 | Fsp2 | Fsp3 | Fsp4 | Fsp5 | Fsp1 | Fsp2 | Fsp3 | Fsp4 | Fsp5 | Fsp8 | Fsp1 | Fsp2 | Fsp3 | Fsp4 | Fsp5 | Fsp1 | Fsp2 | Fsp3 | Fsp4 | Fsp5 | Fsp6 | Fsp7 |
| **Mammals** | | | | | | | | | | | | | | | | | | | | | | | | | |
| **Artiodactyla** | | **295** | | | | | | | | | | | | | | | | | | | | | | | |
| Bovidae | Cephalophus silvicultor | 65 | 1 | | 4 | 3 | 8 | 1 | 1 | 7 | 1 | 14 | | 2 | 2 | 1 | | 2 | 3 | 3 | 4 | 1 | 3 | | |
| | kobus ellipsiprymnus | 4 | | | | | 3 | | | | | 1 | | | | | | | | | | | | | |
| | Syncerus caffer | 126 | 3 | | 5 | | 7 | 1 | 1 | 3 | 2 | 9 | | 2 | 1 | 10 | 5 | 7 | 1 | 2 | 8 | 1 | 6 | | 1 |
| | Tragelaphus spekii | 95 | | | | | | | | 1 | | 6 | | 4 | 1 | 5 | 1 | 3 | 7 | 6 | 9 | 4 | 12 | 1 | |
| Hippopotamidae | Hippopotamus amphibius | 2 | | | | | | | | | | 1 | | | | | | | | | | | | | |
| Suidae | Potamochoerus porcus | 3 | | | | | | | | | | 2 | | | | | | | | | | | | | |
| **Carnivora** | | **1** | | | | | | | | | | | | | | | | | | | | | | | |
| Herpestidae | Herpestinae sp | 1 | | | | | 1 | | | | | | | | | | | | | | | | | | |
| **Primates** | | **67** | | | | | | | | | | | | | | | | | | | | | | | |
| Hominidae | Gorilla gorilla | 3 | | | | | 2 | | | | | | | | | | | | | | | | | | |
| | Homo sapiens | 64 | | | 1 | 1 | 13 | | | 2 | | 22 | 1 | 3 | | 1 | 1 | 2 | | 1 | | | 4 | | 1 |
| **Proboscidae** | | **10** | | | | | | | | | | | | | | | | | | | | | | | |
| Elephantidae | Loxodonta cyclotis | 10 | | | | | | | | | | 7 | | | | | | 2 | | | | | | | |
| **Reptiles** | | | | | | | | | | | | | | | | | | | | | | | | | |
| **Crocodilia** | | **23** | | | | | | | | | | | | | | | | | | | | | | | |
| Crocodylidae | Crocodylus niloticus | 3 | | | | | 3 | | | | | | | | | | | | | | | | | | |
| | Mecistops cataphractus | 19 | | | 1 | | 1 | | | | | 6 | | | | | | | | | | | | | |
| | Osteolaemus tetraspis | 1 | | | | | | | | | | 1 | | | | | | | | | | | | | |
| **Squamata** | | **12** | | | | | | | | | | | | | | | | | | | | | | | |
| Pythonidae | Python sebae | 8 | | | | | 2 | | | | | 2 | | | | | | | | | | | | | |
| Varanidae | Varanus sp | 4 | | | | | | | | 2 | | 1 | | | | 1 | | | | | | | | | |
| **Testudines** | | **16** | | | | | | | | | | | | | | | | | | | | | | | |
| Testunidae | Kinixys erosa | 1 | | | | | | | | | | 1 | | | | | | | | | | | | | |
| Pelomedusidae | Pelusios castaneus | 3 | | | | | | | | 1 | 1 | 1 | | | | | | | | | | | | | |
| | Pelusios chapini | 1 | | | | | | | | | | 1 | | | | | | | | | | | | | |
| | Pelusios marani | 11 | | | | | 3 | | | 3 | | 8 | | | | | | | | | | | | | |

*Table 2 continued*

## Number of identified blood meals by fly species (Fsp)

| | Total | Moukalaba-Doudou Rainy season | Moukalaba-Doudou Dry season | Lopé Rainy season | Lopé Dry season |
|---|---|---|---|---|---|
| **Birds** | | | | | |
| **Ciconiiformes** | 4 | | | | |
| Ciconiidae  *Ciconia sp* | 4 | | 1 | 2 | |
| **8 orders/12 families  20 species** | 428 | 3 / 1 / 11 / 4 | 41 / 2 / 2 / 16 / 4 | 89 / 1 / 6 / 7 / 18 / 7 | 16 / 11 / 11 / 22 / 6 / 25 / 1 / 2 |

Fsp1 = *Glossina caliginea*; Fsp2 = *G. fusca congolensis*; Fsp3 = *G. fuscipes fuscipes*; Fsp4 = *G. pallicera newsteadi*; Fsp5 = *G. palpalis palpalis*; Fsp6 = *Stomoxys calcitrans*; Fsp7 = *S. niger niger*; Fsp8 = *Chrysops* sp.

**Table 3.** Number and origin of blood meals according to the fly species (Fsp), park and climatic season.

**Number of identified blood meals by fly species (Fsp)**

| Taxonomic group/Order/Family | Host species | N° Identified | La Lékédi Rainy season Fsp1 | Fsp3 | Fsp4 | Fsp5 | Fsp6 | La Lékédi Dry season Fsp1 | Fsp2 | Fsp3 | Fsp4 | Fsp5 | Ivindo Dry season Fsp1 | Fsp2 | Fsp3 | Fsp5 |
|---|---|---|---|---|---|---|---|---|---|---|---|---|---|---|---|---|
| **Mammals** | | | | | | | | | | | | | | | | |
| **Artiodactyla** | | **295** | | | | | | | | | | | | | | |
| Bovidae | Cephalophus silvicultor | 65 | | 2 | 1 | 3 | | | | | | | | | | |
| | kobus ellipsiprymnus | 4 | | | | | | | | | | | | | | |
| | Syncerus caffer | 126 | 2 | 1 | | 1 | | 6 | 10 | 8 | 8 | 13 | 1 | | 1 | 1 |
| | Tragelaphus spekii | 95 | | 1 | | 3 | 1 | 2 | 8 | 5 | 5 | 9 | | 1 | | |
| Hippopotamidae | Hippopotamus amphibius | 2 | | | | | | | | | | | | | | |
| Suidae | Potamochoerus porcus | 3 | | | | | | | | | | | | | | |
| **Carnivora** | | **1** | | | | | | | | | | | | | | |
| Herpestidae | Herpestinae sp | 1 | | | | | | | | | | | | | | |
| **Primates** | | **67** | | | | | | | | | | | | | | |
| Hominidae | Gorilla gorilla | 3 | | | | | | | | | | | | | | |
| | Homo sapiens | 64 | | | | 4 | | | | 1 | | 5 | | | | |
| **Proboscidea** | | **10** | | | | | | | | | | | | | | |
| Elephantidae | Loxodonta cyclotis | 10 | | | | | | | | | | | | | | |
| **Reptiles** | | | | | | | | | | | | | | | | |
| **Crocodilia** | | **23** | | | | | | | | | | | | | | |
| Crocodylidae | Crocodylus niloticus | 3 | | | | | | | | | | | | | | |
| | Mecistops cataphractus | 19 | | | | | | 2 | 2 | 4 | 2 | 3 | | | | |
| | Osteolaemus tetraspis | 1 | | | | | | | | | | | | | | |
| **Squamata** | | **12** | | | | | | | | | | | | | | |
| Pythonidae | Python sebae | 8 | | | | 2 | | | | 2 | | 2 | | | | |
| Varanidae | Varanus sp | 4 | | | | | | | | | | | | | | |
| **Testudines** | | **16** | | | | | | | | | | | | | | |
| Testunidae | Kinixys erosa | 1 | | | | | | | | | | | | | | |
| Pelomedusidae | Pelusios castaneus | 3 | | | | | | | | | | | | | | |
| | Pelusios chapini | 1 | | | | | | | | | | | | | | |
| | Pelusios marani | 11 | | | | | | | | | | | | | | |
| **Birds** | | | | | | | | | | | | | | | | |

*Table 3 continued*

**Number of identified blood meals by fly species (Fsp)**

|  |  | Total | La Lékédi | | | | | | | | Ivindo | | |
|---|---|---|---|---|---|---|---|---|---|---|---|---|---|
|  |  |  | Rainy season | | | Dry season | | | | | Dry season | | |
| **Ciconiiformes** |  | 4 |  |  |  |  |  |  |  |  |  |  |  |
| Ciconiidae | *Ciconia sp* | 4 | 2 | 4 | 1 | 13 | 1 | 8 | 20 | 20 | 15 | 32 | 1 |
| **8 orders/12 families** | **20 species** | **428** |  |  |  |  |  |  |  |  | 1 | 2 | 2 |

Fsp1 = *Glossina caliginea*; Fsp2 = *G. fusca congolensis*; Fsp3 = *G. fuscipes fuscipes*; Fsp4 = *G. pallicera newsteadi*; Fsp5 = *G. palpalis palpalis*; Fsp6 = *Stomoxys calcitrans*; Fsp7 = *S. niger*, Fsp8 = *Chrysops sp.*

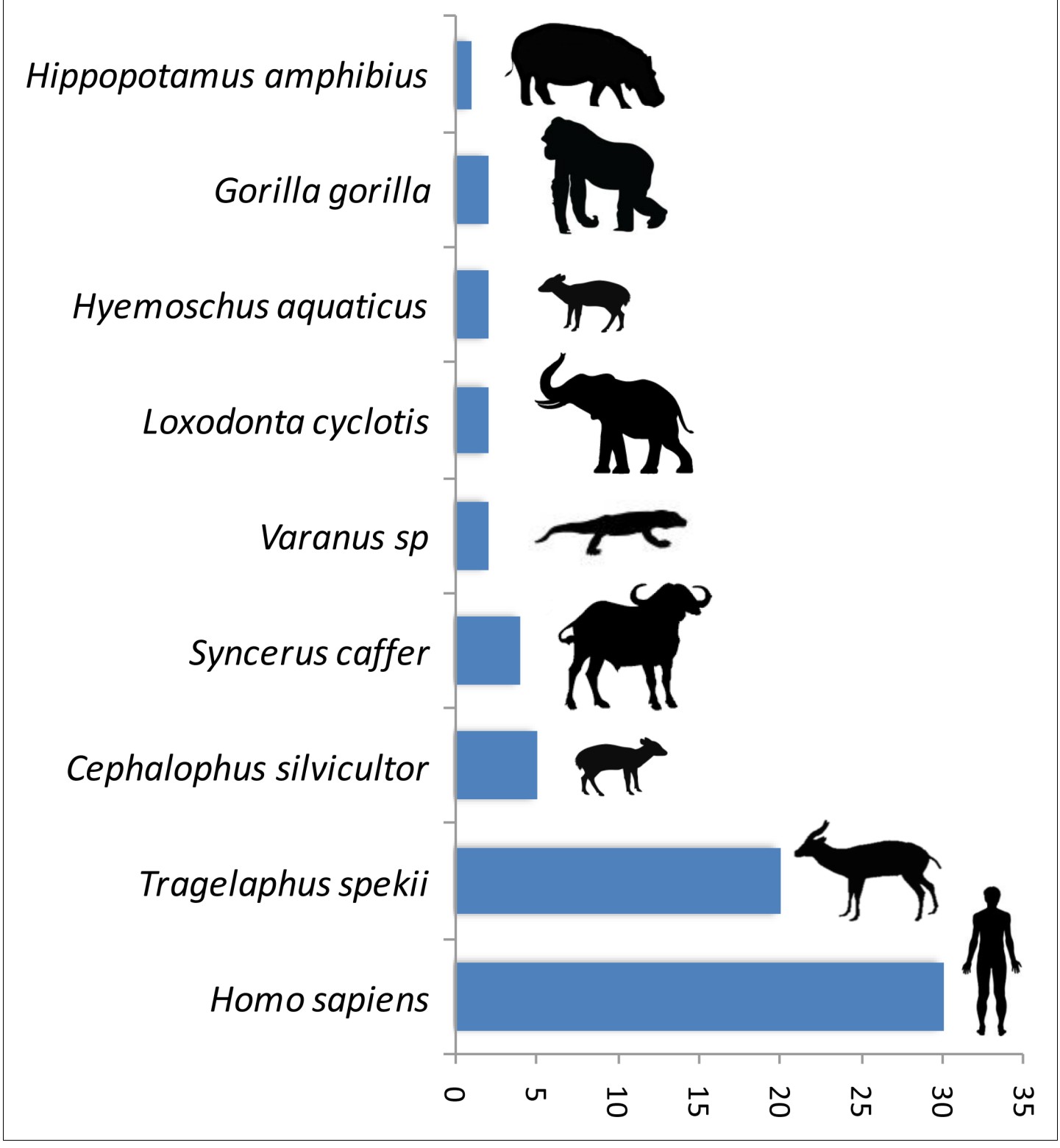

**Figure 2.** Number of blood meals identified using the shorter PCR system of *Boessenkool et al. (2012)* out of the previously unidentified 89 blood meals.

described lineages. In addition, the method allowed identifying the natural hosts of parasites for which only the vectors were known.

**Table 4.** *Cytb* sequences of parasites recovered in this study and of those used as references for phylogenetic analyses and their Genbank accession numbers.

| Isolate | Accession number |
| --- | --- |
| *Anopheles coustani* | KT367855 |
| *An. gabonensis2* | KT367852 |
| *An. gabonensis279* | KT367853 |
| *An. gabonensis3* | KT367861 |
| *An. marshallii* | KT367857 |
| *An. moucheti* | KT367864 |
| *An. obscurus2* | KT367846 |
| *An. obscurus78* | KT367849 |
| *Cephalophus_silvicultor_1336* | KY631949 |
| *Cephalophus_silvicultor_1368* | KY631947 |
| *Cephalophus_silvicultor_484* | KY631963 |
| *Ciconia_sp_445* | KY631985 |
| *E15_Podocnemis_expansa_Peru* | KF049492 |
| *E24_Podocnemis_expansa_Peru* | KF049495 |
| *Gorilla_gorilla_34* | KY631983 |
| *Gorilla_gorilla_756* | KY631982 |
| *Gorilla_gorilla_761* | KY631981 |
| *P_sp._JA7_J725* | GU252027 |
| *Haemoproteus_majoris* | AY099045 |
| *Haemoproteus_sp.* | HM222472 |
| *Haemoproteus_sp._GA02CI1* | HM222486 |
| *Haemoproteus_sp._NA16K65* | HM222487 |
| *Hepatocystis_sp. AA201_blike* | JQ070951 |
| *Hepatocystis_sp.* | JQ070884 |
| *Hepatocystis_sp._AA2012* | JQ070956 |
| *HO11_Cephalophus_nigrofons* | KT367819 |
| *HO13_Cephalophus_monticola* | KT367833 |
| *HO613_Cephalophus_monticola* | KT367836 |
| *HO9_Kinixys_erosa* | KT367843 |
| *Homo_sapiens_476* | KY631978 |
| *Homo_sapiens_481* | KY631977 |
| *Homo_sapiens_57* | KY631979 |
| *Homo_sapiens_574* | KY631976 |
| *Homo_sapiens_635* | KY631975 |
| *Homo_sapiens_636* | KY631974 |
| *Homo_sapiens_638* | KY631973 |
| *Homo_sapiens_639* | KY631972 |
| *Homo_sapiens_668* | KY631969 |
| *Homo_sapiens_806* | KY631968 |
| *Homo_sapiens_832* | KY631967 |
| *Leucocytozoon_caulleryi* | AB302215 |
| *Leucocytozoon_dubreuli* | AY099063 |
| *Leucocytozoon_majoris* | FJ168563 |

*Table 4 continued on next page*

*Table 4 continued*

| Isolate | Accession number |
| --- | --- |
| *Leucocytozoon_sabrazesi* | AB299369 |
| *M0278_Cephalophus_monticola* | KT367834 |
| *NG238_Kinixys_erosa* | KT367844 |
| *NG277_Ceratogymna_atrata* | KT367825 |
| *NY195_Cephalophus_dorsalis* | KT367838 |
| *Nycteria_sp._R_alc_C9_1* | KF159720 |
| *Nycteria_sp._R_lan_G3_1* | KF159690 |
| *OI52_Pangolin* | KT367818 |
| *OL123_Cephalophus_monticola* | KT367822 |
| *OL131_Cephalophus_callipygus* | KT367830 |
| *P_adleri* | HM235081 |
| *P_azurophilum* | AY099055 |
| *P_billcollinsi* | KP875474 |
| *P_blacklocki* | HM235065 |
| *P_cynomolgi* | AB444126 |
| *P_falciparum_3D7* | AF069605 |
| *P_gaboni* | JF895307 |
| *P_gallinaceum* | AF069612 |
| *P_gonderi* | JF923751 |
| *P_knowlesi* | JQ345504 |
| *P_malariae* | HM000110 |
| *P_ovale* | GU723548 |
| *P_praefalciparum_MOEB* | JF923761 |
| *P_reichenowi* | KP875479 |
| *P_relictum* | AY733090 |
| *P_sp._DAJ* | JF923753 |
| *P_vivax* | KF591834 |
| *P_atheruri* | AY099054 |
| *P_giganteum* | AY099053 |
| *P_vinckei_isolate_1* | KJ700853 |
| *P_vinckei_isolate_2* | KJ700854 |
| *P_yoelii_killicki* | DQ414658 |
| *P. atheruri* | HQ712051 |
| *P. cyclopsi_Hip_cy_L4_1_Schaer* | KF159674 |
| *P. voltaicum_M_ang_G1_1_1* | KF159671 |
| *Parahaemoproteus_sp._bird_sp.17* | GQ141581 |
| *Parahaemoproteus_sp. _bird_sp.19* | GQ141585 |
| *Parahaemoproteus_vireonis* | FJ168561 |
| *Plasmodium_sp._bird* | GQ141574 |
| *Plasmodium_sp._bird_sp._12* | HM222485 |
| *Plasmodium_sp._GD2_GD201* | GU252012 |
| *Plasmodium_sp._lineage_JA01* | KM598212 |
| *Polychromophilus_melanipherus_haplotype_VIII* | KJ131277 |
| *Polychromophilus_murinus_haplotype_3* | HM055585 |

*Table 4 continued on next page*

*Table 4 continued*

| Isolate | Accession number |
| --- | --- |
| Polychromophilus_sp._Min_vil_G3_2 | KF159699 |
| Polychromophilus_sp._Pip_gran_G3_1 | KF159714 |
| Polychromophilus_sp._Neo_cap_G3 | KF159700 |
| Syncerus_caffer_1138 | KY631953 |
| Syncerus_caffer_1417 | KY631942 |
| Tragelaphus_eurycerus_1324 | KY631950 |
| Tragelaphus_spekii_1051 | KY631961 |
| Tragelaphus_spekii_1155 | KY631959 |
| Tragelaphus_spekii_1175 | KY631958 |
| Tragelaphus_spekii_1228 | KY631957 |
| Tragelaphus_spekii_1245 | KY631956 |
| Tragelaphus_spekii_1291 | KY631955 |
| Tragelaphus_spekii_1299 | KY631954 |
| Tragelaphus_spekii_1300 | KY631952 |
| Tragelaphus_spekii_1306 | KY631951 |
| Tragelaphus_spekii_1348 | KY631948 |
| Tragelaphus_spekii_1386 | KY631946 |
| Tragelaphus_spekii_1394 | KY631945 |
| Tragelaphus_spekii_1399 | KY631944 |
| Tragelaphus_spekii_1413 | KY631943 |
| Tragelaphus_spekii_385 | KY631964 |
| Tragelaphus_spekii_56 | KY631965 |
| U65_Podocnemis_unifilis_Peru | KF049506 |
| Unknown_host_1036 | KY631960 |
| Unknown_host_110 | KY631966 |
| Unknown_host_512 | KY631962 |
| Unknown_host_520 | KY631980 |
| Unknown_host_649 | KY631971 |
| Unknown_host_665 | KY631970 |
| Unknown_host_819 | KY631984 |

Concerning the method efficiency, 30% of blood meals were obtained from 4099 hematophagous flies. This result is consistent with previous studies (*Mavoungou et al., 2008*; *Simo et al., 2012*) showing that most hematophagous flies caught using traps are often seeking hosts for a blood meal. Other methods using a dip net seem to have a better capture efficiency with more than 40% of engorged flies caught on their resting places (*Gouteux et al., 1984*). However, this method requires spending a lot of time in the field because of difficulties in finding their resting sites and catching the flies.

Tsetse flies provided 99% of the collected blood meals (54% by *Glossina palpalis palpalis*) and they are an interesting candidate as 'flying syringes'. Indeed, differently from stomoxids and tabanids, both sexes are exclusively hematophagous in tsetse flies. In addition, *G. p. palpalis* is considered to be an opportunistic species concerning its feeding behaviour, thus explaining the large diversity of blood meals (*Clausen et al., 1998*; *Simo et al., 2008*; *Weitz, 1963*). Conversely, stomoxids and tabanids show sex-specific differences in feeding behaviour and this may partly explain the smaller number of blood meals collected in these two families. In stomoxids, both sexes are

hematophagous, but males sometimes feed on nectar (*Wall and Shearer, 1997*). Moreover, the digestion of stomoxids starts more rapidly than in the other hematophagous flies (*Moffatt et al., 1995*). Male and female tabanids feed on nectar just after their emergence as adults. Only after having been fertilized, females start sucking blood (*Mullens, 2002*). Therefore, engorged stomoxid and tabanid flies are more difficult to capture. Additionally, the lack of engorged stomoxids and tabanids could be explained by the fact that we sampled flies only at floor level. Indeed, some stomoxid species readily feed on arboreal monkeys that are mostly found higher in the tree layer (*Mavoungou et al., 2008*).

The low rate (35%) of blood meal identifications could be explained by the degradation of host DNA during digestion in the fly midgut or by a too small blood quantity in the midgut. The stage of digestion might influence DNA degradation and the host identification efficiency. Nevertheless, the diversity of hosts we successfully identified, mainly in tsetse fly blood meals, was large, including big terrestrial (elephants) and semi-aquatic mammals (hippopotamus) and also reptiles and birds. As previously noted, the diversity of blood meals can be due to the fly high mobility, their opportunistic feeding behaviour and their frequent feeding. In our study, most blood meals were from terrestrial animals (i.e. that live primarily on the ground) and very few from arboreal species. As mentioned above, this result is potentially biased by the trophic preferences of tsetse flies and by the capture method that excluded canopy levels. Previous studies have shown that hematophagous flies sampled in canopies mainly feed on arboreal species (*Mavoungou et al., 2008*). Therefore, changes in trap position could broaden the range of host species analysed. We can also notice the absence of small mammals (e.g., rodents or bats) within the diversity of host vertebrates we identified. This may be explained by the trophic preferences of the flies we sampled which could have a preferential taste for large vertebrates as previously documented for tsetse flies (e.g. [*Muturi et al., 2011*; *Späth, 2000*]).

Concerning pathogen detection, we detected extant haemosporidian parasites in 8.65% of the 428 blood meals for which the host origin was successfully identified. Moreover, we also detected parasites in blood meals of unknown origin, thus increasing the number of detected parasites. Together, these results show that blood meals collected from hematophagous flies are suitable for tracking blood-borne pathogens from wild animals. Haemosporidian pathogens ingested by hematophagous flies during their blood meal can remain detectable in the fly digestive tract even after partial digestion of the blood meal. We observed congruence between the identified hosts and the detected pathogens. As expected, *P. falciparum* was detected in human blood and *P. adleri* in gorilla blood. Haemosporidian lineages are often host-specific or restricted to certain classes of vertebrate hosts. Therefore, the unknown host could be inferred from the detected haemosporidian species (*Figure 1c*). For example, the blood meal from unknown host N°110 could have originated from a *Kinixys* turtle (*Kinixys sp.*). Similarly, the blood meals from the unknown hosts N°649, 520, 665, 512 and 819 could have originated from humans (*Homo sapiens*).

The present study demonstrates the possibility to use hematophagous flies as 'flying syringes' to analyze the diversity of pathogens circulating in wildlife. We think that there is now room for improvement of the tool; for instance, by improving the methods used to identify the blood meals and the pathogens. Since DNA is likely to be degraded in many blood meals (*Calvignac-Spencer et al., 2013*; *Schnell et al., 2012*), the use of PCR systems targeting fragments of shorter size could potentially improve the performance of detection. A trial study based on 89 previously unidentified blood meals using a PCR system amplifying a shorter fragment (<150 bp) (*Boessenkool et al., 2012*) than the one used in the present study allowed the identification of 76% (n = 68) of the hosts (*Figure 2*). This represents an important gain of sensitivity. However, these primers are still not ideal for our purpose as they were designed for optimal amplification of mammal DNA and often fail to properly amplify the DNA of other classes of vertebrates. A similar PCR system targeting the entire range of vertebrates still remains to be developed. For *Plasmodium*, our trial for amplifying a shorter fragment of *Cytb* (<200 bp) using a combination of previously published primers did not increase the sensitivity. Indeed, out of 91 samples for which the blood meal was successfully identified but in which no haemosporidian infection was detected with our long *Cytb* PCR system, only one was shown to be positive with the short PCR system. However, it is possible that other PCR systems, more optimized, could indeed improve the sensitivity of *Plasmodium* detection. Another direction of improvement could be the use of high-throughput sequencing technologies on pools of blood-engorged flies or amplicons to ease the identification of both hosts and parasites (especially

in the case of mixed blood meals or mixed infections). Finally, another way to improve the tool could be to use high-throughput multiplexed pathogen detection methods for the simultaneous testing of many samples in rapid succession. With such improvements, this approach of 'xenorsurveillance' could usefully complete recently developed methods based on the analysis of other invertebrates (carrion flies (*Hoffmann et al., 2016*), mosquitoes [*Grubaugh et al., 2015*]) and become an innovative way for the concomitant surveillance of many enzootic blood-borne pathogens, such as viruses (chikungunya, Zika), bacteria, protozoa and macro-parasites. The use of hematophagous flies as 'flying syringes' could indeed improve public health management by allowing the surveillance and early detection of zoonotic pathogens and thus prevent they spread to humans before they cause massive infections. This tool could also help to better understand the circulation in wildlife of other enzootic viruses, such as chikungunya or Zika, especially at the interface between natural/sylvan environments and, consequently, improving our knowledge of their natural history. From a broader perspective, this method could also be useful for people interested in wildlife biodiversity and conservation. Indeed, it could help monitoring the wildlife diversity within a specific region as demonstrated with other invertebrate systems (*Calvignac-Spencer et al., 2013*; *Lee et al., 2015*; *Schnell et al., 2012*; *Schubert et al., 2015*). More importantly, it could also allow detecting the emergence of new diseases in wild animals that may threaten their long-term survival.

## Conclusion

Despite the significant scientific advances in the medical field, humans are still unable to predict where, when and how epidemics arise. Around 60% of emerging diseases in humans are of zoonotic origin. The progressive reduction of wild habitats will increase the contacts between humans and species that are potential reservoirs of diseases. We propose here a new non-invasive tool that can help identifying pathogens that circulate in wildlife before they spread in humans.

## Materials and methods

### Study sites

The fly sampling was carried out in four wildlife reserves in Gabon (*Figure 1a*): Moukalaba-Doudou National Park (MDNP; S: 2° 26′ 08″/E: 10° 25′ 18′′), La Lopé National Park (LNP; S: 0° 31′ 31″/E: 11° 32′ 34″), La Lékédi Park (LP; S: 1° 45′ 32″/E: 13° 03′ 16″) and Ivindo National Park (INP; N: 0° 30′ 82″/E: 12° 48′ 20″). Both MDNP and LNP are dominated by mature forests and mosaic forest-savannah. The INP is largely dominated by mature forest with some open biotopes that characterize the secondary forest. The LP is a private park dominated by large savannahs and some secondary forest and primary forest patches.

### Sampling strategy

Hematophagous flies were sampled during the rainy and dry seasons between 2012 and 2014. In INP and MDNP, sampling was done during two years following a gradient of human activity from primary forest to villages. In the other parks, flies were sampled during a single year. Flies were collected by using Vavoua and Nzi traps (*Laveissiere and Grebaut, 1990*; *Acapovi et al., 2001*; *Mihok, 2002*; *Gilles et al., 2007*). The Vavoua trap, initially developed for the capture of tsetse flies was also successfully used for the capture of stomoxids at La Réunion Island (*Laveissiere and Grebaut, 1990*; *Gilles et al., 2007*). The Nzi trap was more adapted to the capture of *Glossina pallidipes* and tabanids in Africa (*Acapovi et al., 2001*; *Mihok, 2002*). In each park, we placed 24 traps (12 Vavoua and 12 Nzi) during 2 weeks per climatic season. Each trap was activated from 7:00 AM to 5:00 PM.

### Identification and dissection of hematophagous flies

Freshly collected hematophagous flies were identified using a stereo-microscope and taxonomic procedure. The fly species (tsetse, stomoxids and tabanids) was determined following the determination keys of *Pollock (1982)*, *Brunhes et al., 1998*, *Zumpt, 1973*, *Garros et al. (2004)* and *Oldroyd (1973)*, on the basis of their morphological characteristics, such as size, color, wing venation structure and proboscis.

After species identification, engorged flies were dissected individually in a drop of Dulbecco's phosphate buffered saline solution (1x DPBS) to isolate blood meals from midgut. Each hematophagous fly was dissected on a slide using one forceps and one scalpel that were changed each time to avoid contaminations. Each blood meal was transferred in a 1.5-ml microtube containing 50 µl of RNA*later* stabilization solution (Qiagen: Store at RT Tissue Collection) to stabilize and protect nucleic acids of vertebrate hosts and pathogens contained in the blood meals. Samples were kept at ambient temperature during field session and then frozen at −80°C until DNA extraction.

## DNA extraction

Samples were centrifuged at 15,000 rpm at 4°C for 10 min to remove the RNA*later* solution. Pellets were used to extract DNA using the DNeasy Blood and Tissue Kit (Qiagen) according to the manufacturer's instructions. Extracted DNA was eluted in 100 µl of buffer AE and stored at −20°C.

## Blood meal identification

The origin of blood meals was determined using the extracted DNA to amplify a 450 bp fragment of the *cytochrome b* (*Cytb*) gene using previously published primers (*Townzen et al., 2008*). PCR amplifications were performed using a GeneAmp 9700 thermal cycler (Applied Biosystems, USA) with 50 µL reaction mixtures containing 4 µL template DNA, 10 mM Tris-HCl (pH = 9), 50 mM KCl, 3 mM MgCl$_2$, 20 pmol each primer (5'CCCCTCAGAATGATATTTGTCCTCA3' and 5'CCATCCAACATC TCAGCATGATGAAA3'), 200 mM dNTP and 1 U Taq polymerase. The thermal cycling conditions consisted of 3.5 min at 95°C, 40 cycles of 30s at 95°C, 50s at 58°C, and 40s at 72°C, followed by 5 min at 72°C. When *Cytb* amplification failed, a 330 bp and/or a 660 bp fragment of the *cytochrome oxydase* subunit I (*COI*) gene was amplified using previously described primers and protocols (*Townzen et al., 2008*). All PCR-amplified products (10 µl) were run on 1.5% agarose gels in TBE buffer, and positive samples were sent to Beckman Coulter Genomics (France) for sequencing in both directions (forward and reverse) after purification. Consensus sequences were compared with existent sequences using the NCBI nucleotide Blast search (*Altschul et al., 1990*) to determine the host species. Hosts were identified when the amplified and reference sequences showed at least 98% similarity.

## Haemosporidia detection and identification

Haemosporidian parasite detection was performed in samples with identified blood meal origin and also in 100 randomly chosen samples for which blood meal origin could not be identified.

Haemosporidian parasites were detected by PCR amplification of a portion of the *Cytb* gene (~790 bp) using a nested PCR protocol, as previously published (*Ollomo et al., 2009*). PCR products were checked on 1.5% agarose gels before shipment to EUROFINS MWG (Germany) for sequencing in both directions (reverse and forward) after purification. Multiple alignments of haemosporidian sequences were done using Muscle (*Edgar, 2004*). A phylogenetic tree with the haemosporidian sequences obtained in our study and a set of reference sequences was built using Maximum likelihood (ML) methods and phylogeny.fr (*Dereeper et al., 2008*) (see *Table 4* for accession numbers). The ML model used for construction of the tree was GTR (General Time Reversible)+Γ (Gamma distribution)+I (Invariable site distribution).

## Anti-contamination procedures

Several measures were taken to avoid contaminations during our manipulations. Extraction of DNA was performed at the CIRMF (Gabon) in a laboratory working on mosquitoes. The room in which extraction was performed was away from the rooms in which DNA was amplified in this lab.

DNA extracts were then sent to France at the IRD (Montpellier). There, blood meal and *Plasmodium* identification was performed. This lab had never worked before on *Plasmodium* from ungulates or reptiles. Amplification of host DNA was never or very rarely performed in this lab. When the work was performed, no work on *Plasmodium* has been performed in this lab for almost 4 years. In addition, the laboratory is designed to avoid contaminations. Clearly defined and separated areas are devoted for each step of the PCR process: one area is devoted to the preparation of reagents (mix PCR). Another room is dedicated to the pre-PCR manipulation (loading of native DNA). This step is done under a cabinet to avoid contamination of the sample with DNA from the operator. Finally, an

area is devoted to PCR-amplified DNA. In this area, cabinets are used to deposit the first PCR product into the reagents of the second PCR (for nested PCRs). All cabinets are equipped with UV lamps and are always decontaminated with DNA-free solutions before and after manipulations. Gloves and coats are changed when moving between the areas and plugged tips are used at all steps. Blank controls were always incorporated at all steps of the experimental procedure and were always negative.

Several observations confirm the authenticity of our results: (1) >80% of the hosts that were found have never been manipulated in our lab (hosts that are not humans or non-human primates); (2) the parasite always corresponded to the expected host (antelope parasites were always found in antelopes, human parasites in humans and gorilla parasites in gorillas). Contaminations by external DNA would have lead to random association of hosts and parasites; (3) A new lineage of parasites was discovered.

## Trial study to amplify shorter PCR fragments

Since DNA is likely to be degraded in many of our samples, the use of PCR systems targeting fragments of shorter size might improve performance. To determine if this could be the case with our study system, we performed supplementary analyses using (1) a PCR system targeting a shorter fragment of the vertebrate mitochondrial DNA to identify the blood meal origin and (2) a PCR system targeting a shorter fragment of the *Cytb* DNA to identify the parasite. For the identification of the host, the PCR system used was the one amplifying a fragment of 150 bp of 16S as described in *Boessenkool et al. (2012)* and using the primers 16Smam1 and 16Smam2. This PCR system was used on blood meals that failed to be identified using our original PCR system (see the paragraph 'Blood meal identification'). A total of 89 blood meals were tested for this trial study. For the parasite, we designed new primer sets to amplify a shorter fragment of the *Cytb* gene of the parasite (~177 bp). This new PCR system was applied to blood meals for which the host was identified but that were negative to *Plasmodium* with our long PCR system (~790 bp, see Material and methods above). A total of 91 blood samples were tested. For the first round of amplifications, we used 6 µL of DNA template in a 25 µL reaction volume, containing: 12.5 µL of Mix PCR (Qiagen), 2.5 µL solution Q (Qiagen), and 4 pmol of each primer (cytb1F CTCTATTAATTTAGTTAAAGCACACTT and 454R CCWGTWGCYTGCATYTATCT). Cycling conditions were 15 min at 95°C, 30 s at 94°C, 90 s at 57°C, 90 s at 72°C (40 cycles), and 10 min at 72°C. For the second round of amplification, we used 1.5 µL of the first PCR template in a 25 µL reaction volume, containing 2.5 µL of 10× buffer, 1.25 mM MgCl$_2$, 250 µM of each dNTP, 10 pmol of each primer (454F2 WAATTAYCCATGYCCATTRAA and Plas1rc CACCATCCACTCCATAATTCTC), and 0.1 unit Taq Platinum (Invitrogen). Cycling conditions for the second round were 5 min at 95°C, 30 s at 94°C, 30 s at 50°C, 90 s at 72°C (35 cycles), and 10 min at 72°C. The amplified products (5 µL) were run on 1.5% agarose gels in TAE buffer. The PCR-amplified products (177 bp) were used as templates for sequencing. DNA sequencing was performed by Eurofins MWG.

## Acknowledgements

Authors thank all the reviewers for their constructive and helpfull comments. This study was carried out with a financial support of: 'Agence Universitaire de la Francophonie' (AUF), the 'Service de Cooperation et d'Action Culturelle' (SCAC) of French Embassy in Gabon, the 'Institut Français' of Libreville (IF), the 'Conseil Régional de Bourgogne' and the 'Bonus Qualité Recherche' (BQR) of Université de Bourgogne. This work was also funded by Institut de Recherche pour le Développement (Laboratoire Mixte International ZOFAC), Centre International de Recherches Médicales de Franceville (CIRMF), as well as the Agence Nationale de la Recherche (ANR) programme Jeunes Chercheuses Jeunes Chercheurs (JCJC) Sciences de la Vie, de la Santé et des Ecosystèmes 7–2012 project ORIGIN (ANR JCJC SVSE 7–2012 ORIGIN). We thank the Agence Nationale des Parcs Nationaux (ANPN) and the Centre National de la Recherche Scientifique et Technologique (CENAREST) of Gabon who authorized this study and facilitated the access to national Parks. Authors also thank Eric Willaume from the park of La Lékédi for his help.

# Additional information

## Funding

| Funder | Grant reference number | Author |
|---|---|---|
| Agence Universitaire de la Francophonie | | Paul-Yannick Bitome-Essono<br>Flobert Njiokou<br>Francois Bretagnolle<br>Franck Prugnolle<br>Christophe Paupy |
| Service de Coopération et d'Action Culturelle de l'ambassade de France au Gabon | | Paul-Yannick Bitome-Essono<br>Francois Bretagnolle |
| Laboratoires Mixtes Internationaux | LMI ZOFAC IRD | Benjamin Ollomo<br>Franck Prugnolle<br>Christophe Paupy |
| Centre International de Recherches Médicales de Franceville | | Paul-Yannick Bitome-Essono<br>Benjamin Ollomo<br>Diego Ayala<br>Virginie Rougeron<br>Franck Prugnolle<br>Christophe Paupy |
| Agence Nationale de la Recherche | ANR JCJC 07-2012-ORIGIN | Virginie Rougeron<br>Franck Prugnolle<br>Christophe Paupy |

The funders had no role in study design, data collection and interpretation, or the decision to submit the work for publication.

## Author contributions

P-YB-E, Conceptualization, Methodology, Writing—original draft, Writing—review and editing; BO, Conceptualization, Methodology; CA, NDM, LY-M, A-PO, LB, BM-O, JO-N, PM-N, FN, BM, DA, Investigation, Methodology; PD, Investigation, Writing—review and editing; RW, Formal analysis, Investigation, Methodology; FJA, Investigation, Methodology, Writing—review and editing; FR, Conceptualization, Writing—review and editing; VR, Conceptualization, Investigation, Methodology, Writing—review and editing; FB, FP, CP, Conceptualization, Formal analysis, Supervision, Investigation, Methodology, Writing—original draft, Writing—review and editing

## Author ORCIDs

Franck Prugnolle, http://orcid.org/0000-0001-8519-1253
Christophe Paupy, http://orcid.org/0000-0002-7122-2079

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
