## [Decision Letter]

Thank you for submitting your article "Tracking zoonotic pathogens using blood-sucking flies as "flying syringes"" for consideration by *eLife*. Your article has been favorably evaluated by Prabhat Jha (Senior Editor) and three reviewers, one of whom, Ben Cooper (Reviewer #1), is a member of our Board of Reviewing Editors. The following individual involved in review of your submission has agreed to reveal their identity: Sébastien Calvignac-Spencer (Reviewer #3).

The reviewers have discussed the reviews with one another and the Reviewing Editor has drafted this decision to help you prepare a revised submission.

Summary:

The study demonstrates the potential utility of molecular analysis of blood collected from blood sucking flies for monitoring infectious agents in wild animal sources using as an example an analysis of malaria parasites identified from a collection of 4100 flies (mostly tsetse) captured in pristine habitats in Gabon. From this collection, 1230 flies were engorged with blood and the origin of the blood meal was identified from >400 flies and malaria parasite genetic material from 37 individual flies. A number of know malaria parasite species were identified and a couple of potentially new parasites were found or linked to a host for the first time.

All reviewers thought that this was interesting work and that the kind of approach described in this paper is very promising.

All agreed, however, that a number of essential revisions are required. In particular, it was felt that there was much room for technical improvements and clarifications. For example, the relatively low rate of recovery of vertebrate sequences probably finds an explanation in the excessive length of the PCR system used by the authors – most groups in the eDNA/iDNA field have converged to using systems targeting fragments <200bp (experimental demonstrations of fragment length dependent loss of sensitivity beyond this size can be found in Schnell et al. Curr Biol 2012, Calvignac-Spencer et al. Mol Ecol 2013 for iDNA and a host of eDNA papers).

The reviewers also agreed that individual-based analyses are unlikely to be applied on the massive scales that would be needed to find these needles in this haystack (37 malaria positive out of 4,000 individuals – knowing malaria parasites are hemoparasites!). To demonstrate the potential utility of the approach the authors should ideally show experimentally that technical solutions more in line with high-throughput analyses would work, e.g. no gut dissection + multi-individual pooling + amplicon NGS (and/or combination thereof).

Essential revisions:

1) More analyses should be provided together with additional comparative data from the literature. The authors gave raw data information in their supplementary files, with which pathogen in which blood meal from flies in the different localities. With these, the authors may better investigate the limitations of the method by comparing the relative abundance and diversity of vertebrates with other estimates of wildlife abundance (i.e. census for national parks). They should certainly be able to compare their pathogen results using flies with other studies that have investigated directly the prevalence and diversity of microbes / parasites among the wildlife.

2) Additional lab experiments should be performed to address concerns outlined above:

i) use an alternative, shorter mammal-targeting system on the negatives (e.g. 16S by Taylor 1996, made very popular by Boessenkool et al. 2012);

ii) a shorter Plasmodium targeting system on the mammal positives (unless the system already targeted a fragment in the range of 100-200bp);

iii) a couple of DNA extract pooling experiments coupled with bulk PCR and amplicon NGS (e.g. with 10 pools of 10 DNA extracts or so).

i) and ii) should increase sensitivity (one can expect that given malaria positives in apparently mammal negative flies). iii) will open the way to a more realistic implementation of the tool.

3) The authors should explain very clearly what anti-contamination measures were taken.

The finding of *P. falciparum* and *P. adleri* in a lab famous for their work on great ape malaria parasites will necessarily raise many eyebrows. More generally, the fact that most of the malaria parasite sequences belong to a handful species/lineages also raises the question of potential contaminations, including during the experiments, i.e. PCR products generated during this study that may have contaminated the next experiments. The authors should add a section describing their anti-contamination measures in the Materials and methods.

4) Currently the description and reporting of the statistical analysis is inadequate, and this needs to be addressed:

i) Twice in the subsection “Host identification from blood meals”, p-values are quoted without any accompanying point estimates or confidence intervals. These p-values, on their own, are just about meaningless (an arbitrary difference, of no biological significance, can give a p-value as small as you like with enough data). The authors need to report the effect size and confidence intervals.

ii) Subsection “Data analysis”. This section is too vague. If mixed effects models are used, what is the clustering variable? If a model with binomial distribution was used, what was the link function? How was season modelled? etc. Full details of the models should be given and full results can be reported in the supplementary material.

5) While the reviewers acknowledge that such work has only rarely been performed, it is untrue to state that "the proof of concept has never been obtained". Actually, the concept was even given a name (xenosurveillance) in a paper in which Plasmodium sp. sequences were detected in blood-fed mosquito pools (although the authors rather insisted on their detection of EBV and CDV; Grubaugh et al. PLoS NTD 2015). There are many other papers on this idea that could be cited. Among others:

Calvignac-Spencer S et al. 2013. Carrion fly-derived DNA as a tool for comprehensive and cost-effective assessment of mammalian biodiversity. Molecular Ecology doi: 10.1111/mec.12183

Kent RJ 2009. Molecular methods for arthropod blood meal identification and applications to ecological and vector-borne disease. Molecular Ecology Resources doi: 10.1111/j.1755-0998.2008.02469.x

Lee P-S et al. 2015. Reading Mammal Diversity from Flies: The Persistence Period of Amplifiable Mammal mtDNA in Blowfly Guts (Chrysomya megacephala) and a New DNA Mini-Barcode Target. PLoS ONE 10(4):e0123871. doi:10.1371/journal.pone.0123871

Schubert G et al. 2014. Targeted detection of mammalian species using carrion fly-derived DNA. Molecular Ecology Resources doi: 10.1111/1755-0998.12306and even from blood meals of leeches:

Schnell ID et al. 2015. DNA from terrestrial haematophagous leeches as a wildlife surveying and monitorings tool – prospects, pitfalls and avenues to be developed. Frontiers in Zoology (2015) 12:24

DOI 10.1186/s12983-015-0115-z

Within the last month a closely related paper has also appeared in Scientific Reports "Assessing the feasibility of fly based surveillance of wildlife infectious diseases" doi:10.1038/srep37952. This paper should now be referenced in your revised submission.

---

## [Author Response]

Essential revisions:

1) More analyses should be provided together with additional comparative data from the literature. The authors gave raw data information in their supplementary files, with which pathogen in which blood meal from flies in the different localities. With these, the authors may better investigate the limitations of the method by comparing the relative abundance and diversity of vertebrates with other estimates of wildlife abundance (i.e. census for national parks). They should certainly be able to compare their pathogen results using flies with other studies that have investigated directly the prevalence and diversity of microbes / parasites among the wildlife.

We agree with reviewers that it would be nice if we could compare our results of vertebrate diversity or pathogen prevalence with data from the literature. Unfortunately, these data are lacking for the parks where we worked, preventing a good comparative analysis. The only information we could obtain is, for the best, an incomplete list of mammal species without information on their abundance for the park of La Lopé (White, 1994, Journal of Animal Ecology). What we can notice from the comparison of the list of species found in the blood meals of the flies and the list of large vertebrates present in the park is that there is an overrepresentation of terrestrial vertebrate species and a lack of species found in canopy. There is also a lack of small mammals, like rodents or bats.

For the pathogens, a direct comparison of what has been found with the literature is not possible. Most malaria agents discovered in our study are from antelopes. The only study that was published on it was from bushmeat samples collected all over Gabon, which clearly precludes a comparison with our sites of study as there is a chance that prevalence may greatly vary from one site to another, as previously noted on *Plasmodium/ apes* studies.

2) Additional lab experiments should be performed to address concerns outlined above:

*i) use an alternative, shorter mammal-targeting system on the negatives (e.g. 16S by Taylor 1996, made very popular by Boessenkool et al. 2012);*

*ii) a shorter Plasmodium targeting system on the mammal positives (unless the system already targeted a fragment in the range of 100-200bp);*

*iii) a couple of DNA extract pooling experiments coupled with bulk PCR and amplicon NGS (e.g. with 10 pools of 10 DNA extracts or so).*

i) and ii) should increase sensitivity (one can expect that given malaria positives in apparently mammal negative flies). iii) will open the way to a more realistic implementation of the tool.

Following reviewers’ recommendations, shorter PCR systems were used for 1) the identification of the blood meal and 2) the identification of the infections with haemosporidian parasites.

For the blood meal, we used the primers designed by Taylor et al. 1996 targeting a short fragment of the mitochondrial DNA of mammals. We tested this new PCR on 89 blood meals for which we were not able to identify the host with our previous PCR system. Out of the 89, we were able to identify the host in 76% of the cases after amplification and sequencing. This clearly demonstrates a far better sensitivity of this system based on the amplification of a shorter fragment.

For the parasites, we designed new primers sets (based on previously published primers) to amplify a shorter fragment of the *Cyt-b* gene of the parasite. The newly amplified fragment was shorter than 200 bp. To determine if this would lead to a gain of sensitivity, we tested 91 blood meals for which the host was determined but were negative with previous *Plasmodium* PCR system. We re-did the *Plasmodium* PCR with the new system. We were able to identify one additional positive individual, which do not indicate a large gain in sensitivity for the detection of the parasite. The sensitivity of the two PCRs should however be tested with precision and need more development.

We did not perform any experiment using NGS or high throughput technologies due to a lack of time. However, as highlighted in the Discussion, the tool needs now to be developed and optimized and NGS or the use of high throughput diagnostic method is a clear direction of improvement.

These pilot tests have now been added to the new version of the manuscript.

3) The authors should explain very clearly what anti-contamination measures were taken.

The finding of P. falciparum and P. adleri in a lab famous for their work on great ape malaria parasites will necessarily raise many eyebrows. More generally, the fact that most of the malaria parasite sequences belong to a handful species/lineages also raises the question of potential contaminations, including during the experiments, i.e. PCR products generated during this study that may have contaminated the next experiments. The authors should add a section describing their anti-contamination measures in the Materials and methods.

Several measures were taken to avoid contaminations during our manipulations. Extraction of DNA was performed at the CIRMF (Gabon) in a laboratory working on mosquitoes and not manipulating *Plasmodium* or mammal DNA. The room in which extraction was performed was away from the rooms in which DNA was amplified in this lab.

DNA extracts were then sent in France at the IRD (Montpellier). There, blood meal and *Plasmodium* identification was performed. This lab had never worked before on *Plasmodium* from wild animals, only human and non-human primate *Plasmodium*. Amplification of host DNA was never or very rarely performed in this lab. When the work was performed, no work on *Plasmodium* has been performed in this lab for almost four years. In addition, the laboratory is designed to avoid contaminations. Clearly defined and separated areas are devoted for each step of the PCR process: one area is devoted to the preparation of reagents (mix PCR). Another room is dedicated to the pre-PCR manipulation (loading of native DNA). This step is done under a cabinet to avoid contamination of the sample with DNA from the operator. Finally, an area is devoted to PCR-amplified DNA. In this area, cabinets are used to deposit PCR1 into the reagents of PCR2 (for nested PCRs). All cabinets are equipped with UV lamps and are always decontaminated with DNA free solutions after and before manipulations. Gloves and coats are changed when moving between the areas and plugged tips are used at all steps. Blank controls were always incorporated at all steps of the experimental procedure and were always negative.

Several observations confirm the authenticity of our results: 1) 85% of the hosts that were found were never manipulated in our labs (hosts that are not humans or apes). 2) the parasite always corresponded to the expected host (antelope parasite were always found in antelopes, human parasite in humans and gorilla parasites in gorillas). Contaminations by external DNA would have lead to random association of hosts and parasites. 3) A new lineage of parasites was discovered.

The experimental procedure is entirely detailed in the Materials and methods section of the new version of the manuscript.

4) Currently the description and reporting of the statistical analysis is inadequate, and this needs to be addressed:

*i) Twice in the subsection “Host identification from blood meals”, p-values are quoted without any accompanying point estimates or confidence intervals. These p-values, on their own, are just about meaningless (an arbitrary difference, of no biological significance, can give a p-value as small as you like with enough data). The authors need to report the effect size and confidence intervals.*

*ii) Subsection “Data analysis”. This section is too vague. If mixed effects models are used, what is the clustering variable? If a model with binomial distribution was used, what was the link function? How was season modelled? etc. Full details of the models should be given and full results can be reported in the supplementary material.*

We thank the reviewer for pointing this out and agree. However, we have decided to remove these descriptive statistical analyses because we felt it did not add a lot to the main message of the paper and would bring too much confusion. We therefore deleted Supporting Table S1 from the previous version of the manuscript.

5) While the reviewers acknowledge that such work has only rarely been performed, it is untrue to state that "the proof of concept has never been obtained". Actually, the concept was even given a name (xenosurveillance) in a paper in which Plasmodium sp. sequences were detected in blood-fed mosquito pools (although the authors rather insisted on their detection of EBV and CDV; Grubaugh et al. PLoS NTD 2015). There are many other papers on this idea that could be cited. Among others:

*Calvignac-Spencer S et al. 2013. Carrion fly-derived DNA as a tool for comprehensive and cost-effective assessment of mammalian biodiversity. Molecular Ecology doi: 10.1111/mec.12183*

*Kent RJ 2009. Molecular methods for arthropod blood meal identification and applications to ecological and vector-borne disease. Molecular Ecology Resources doi: 10.1111/j.1755-0998.2008.02469.x*

*Lee P-S et al. 2015. Reading Mammal Diversity from Flies: The Persistence Period of Amplifiable Mammal mtDNA in Blowfly Guts (Chrysomya megacephala) and a New DNA Mini-Barcode Target. PLoS ONE 10(4):e0123871. doi:10.1371/journal.pone.0123871*

*Schubert G et al. 2014. Targeted detection of mammalian species using carrion fly-derived DNA. Molecular Ecology Resources doi: 10.1111/1755-0998.12306and even from blood meals of leeches:*

*Schnell ID et al. 2015. DNA from terrestrial haematophagous leeches as a wildlife surveying and monitorings tool – prospects, pitfalls and avenues to be developed. Frontiers in Zoology (2015) 12:24*

DOI 10.1186/s12983-015-0115-z

Within the last month a closely related paper has also appeared in Scientific Reports "Assessing the feasibility of fly based surveillance of wildlife infectious diseases" doi:10.1038/srep37952. This paper should now be referenced in your revised submission.

We thank reviewers for these references that for the most we indeed knew. There are now cited in the paper. However, most of them concern mammal biodiversity and not pathogen diversity. The aim of our paper is clearly to estimate pathogen diversity which goes a step further to most of these studies cited above. The only two references that are close in principle to what we propose are those of Grubaugh et al. PLos NTD 2015 and the very recent one from Hoffman et al. 2016. Note however that the second reference, although close in principle, uses carrion-flies to detect pathogens from carcasses and not hematophagous flies. There are now cited in the new version of the manuscript and the term “xenosurveillance” is now used.